# Silicon Carbide Precursor: Structure Analysis and Thermal Behavior from Polymer Cross-Linking to Pyrolyzed Ceramics

**Sébastien Vry** [1], **Marilyne Roumanie** [1,*], **Pierre-Alain Bayle** [2], **Sébastien Rolère** [1] **and Guillaume Bernard-Granger** [3]

1   CEA, LITEN, DTNM, Université Grenoble Alpes, 38000 Grenoble, France
2   CEA, IRIG, MEM, Université Grenoble Alpes, 38000 Grenoble, France
3   CEA, DES, ISEC, DMRC, SPTC, LSEM, Université Montpellier, 30207 Marcoule, France
*   Correspondence: marilyne.roumanie@cea.fr

**Abstract:** The Silres H62C methyl-phenyl-vinyl-hydrogen polysiloxane is a promising candidate as a SiC precursor for 3D printing based on photopolymerization reaction. An in-depth nuclear magnetic resonance spectroscopy analysis allowed us to determine its structure and quantify its functional groups. The polysiloxane was found to have a highly branched ladder-like structure, with 21.9, 31.4 and 46.7% of mono-, di- and tri-functional silicon atoms. The polysiloxane cross-links from 180 °C using hydrosilylation between silyl groups (8.4% of the total functional groups) and vinyl groups (12.0%) and contains a non-negligible ethoxy content (2.4%), allowing cross-linking through a hydrolyze/condensation mechanism. After converting the polymer into ceramic and thus releasing mainly hydrogen and methane, the ceramic yield was 72.5%. An X-ray diffraction analysis on the cross-linked and pyrolyzed polysiloxane showed that the ceramic is amorphous at temperatures up to 1200 °C and starts to crystallize from 1200 °C, leading into 3C-SiC carbon-rich ceramic at 1700 °C in an argon atmosphere.

**Keywords:** polymer-derived ceramics; SiC; polysiloxane; NMR characterization

## 1. Introduction

Dense, closed-pore silicon carbide (SiC) is a high-performance material with high mechanical properties (flexural strength of 400–860 GPa), interesting thermal properties (a coefficient of thermal expansion CTE of 4 to $6.10^{-6}$ K$^{-1}$ and thermal conductivity of around 125 W·m$^{-1}$·K$^{-1}$), a low density (3.16 g·cm$^{-3}$) and chemical inertness [1–3]. The main applications are ballistic protection for vehicles and infantry armor or refractory ceramics. SiC is also used in the semiconductor industry (wafer substrates) or as sealing rings, heat exchangers and reactors for the chemical industry and in optical instrumentation (structural parts, telescope mirrors, for example).

Conventionally, SiC-based ceramics are manufactured by powder pressing, followed by sintering. The geometries of parts are consequently limited, and achieving complex shapes/morphologies requires expensive machining. The densification of the ceramic is performed either by pressure-assisted or pressureless sintering, in solid-state or with the contribution of a liquid phase, at temperatures above 2000 °C, using suitable additives (e.g., $Al_2O_3$ and/or $Y_2O_3$, $B_4C$) [4].

The development of 3D printing makes it possible to print SiC parts whose design is not achievable using conventional techniques. Among the various additive manufacturing technologies, SLA (stereolithography apparatus) and DLP (digital light processing) are based on photopolymerization and possess the advantages of low roughness, high accuracy (resolution of ~50 μm) and short cycle duration [5]. The SiC particles strongly absorb the light energy required to activate the photoinitiators in the photosensitive formulation. Therefore, the loading rate of SiC has to be limited to 40 vol.% in the photocurable formulation to print parts [6,7]. He et al. [7] showed that a low loading rate leads to a porous part,

with a relative density of 36%, after debinding and sintering. Thereby, a process of polymer infiltration pyrolysis (PIP), repeated eight times, was performed, with a polycarbosilane as a precursor for the SiC to achieve a relative density of 85%.

Polycarbosilanes and, more specifically, the commercial allyl(methyl)hydrido polycarbosilane (AHPCS) are well-documented polymer-derived ceramics. Many publications detail the synthesis routes, the structures before and after cross-linking of these organic/inorganic polymers using nuclear magnetic resonance (NMR) and the ceramics obtained after heat treatments up to 1600 °C [8–12]. Their attractiveness relies on their ability to lead to SiC at lower temperatures than for conventional powder-based processes and their ease of implementation using various techniques. They are mainly routes to synthesize SiC fibers [13], and few publications are reporting on their printing using DLP [14,15] despite their UV absorption. Nevertheless, polycarbosilanes are expensive and require special handling precautions to limit oxidation and the presence of silica in the final ceramic. On the other hand, polysiloxanes are less expensive and are chemically stable. Martinez-Crespiera et al. [16] used a commercial polysiloxane (RD-684 Polyramic, Starfire Systems Inc), heat-treated at 1100 and 1400 °C in argon, and showed an evolution from an amorphous structure to one built on β-SiC crystallites. Polysiloxanes also appear to have the ability to form SiC at the same temperature as polycarbosilanes. In addition, polysiloxane-based formulations and DLP prints were referenced in the literature, demonstrating the potential of polysiloxanes in photopolymerization additive manufacturing [17–20]. Silres MK, the most studied commercial polysiloxane, is a branched polysiloxane whose structure was investigated using NMR, leading to a high ceramic yield. Zanchetta et al. [20] reported the DLP printability of this solid polysilsesquioxane from a solvent photocurable formulation. Solvent evaporation in a photocurable formulation is complex to manage in a DLP batch, leading to a non-stable formulation under UV irradiation. It would, therefore, be preferable to develop formulations for DLP printing based on liquid polysiloxane with low UV absorption.

The commercial liquid polysiloxane Silres H62C has a low UV absorption in the wavelength range between 365 and 405 nm used for DLP printing. In addition, in a previous investigation, the Silres H62C resin treated at 1700 °C in argon was characterized using transmission electron microscopy (TEM). This characterization showed that the ceramics obtained are made of carbon-enriched SiC [21]. However, this promising polymer for the printing of SiC parts is only rarely referenced alone in the literature and is often associated with Silres MK polysiloxane [22–24].

Silres H62C is described as a methyl-phenyl-vinyl-hydrogen siloxane polymer. However, its linear or branched structure and the proportion of the different functions are not detailed in the literature. The investigations reported in this paper concern the determination of the chemical structure of the Silres H62C polymer and the location of the expected functions using nuclear magnetic resonance (NMR) spectroscopy. Differential scanning calorimetry (DSC) was also used to characterize the Silres H62C cross-linking temperature, while the cross-linking reaction was followed by FT-IR spectroscopy. A thermal gravimetric analysis coupled with a mass spectrometer (TGA-MS) was also used to determine the ceramic yield and identify the released gases during the conversion of the polymer into ceramic. Finally, X-ray diffraction (XRD) was carried out to complete the investigations on the polysiloxane pyrolyzed up to 1700 °C.

## 2. Materials and Methods

### 2.1. Materials

Silres H62C was supplied by Wacker Chemie AG (Germany). It contains a platinum-based thermal catalyst coupled with a radical inhibitor [25]. The polymer comprises vinyl and silyl groups, leading to cross-linking using hydrosilylation/polyaddition. Nevertheless, no precise information on its composition can be found in the literature. Silres H62C is solvent-free and liquid at room temperature, with a viscosity of 1 Pa·s.

### 2.2. NMR Characterization of the Raw Polysiloxane

Several building units can be encountered in polysiloxanes with different chemical shifts in $^{29}$Si NMR spectroscopy (Figure 1). The mono-functional (M) group has a single Si-O bond. The di-functional (D) group, with characteristics of the presence of two oxygen atoms connected to a silicon atom, allows the estimation of the linearity of the polymer. The tri-functional (T) and tetra-functional (Q) groups correspond to Si atoms bound to 3 and 4 oxygen atoms, respectively.

**(a)**

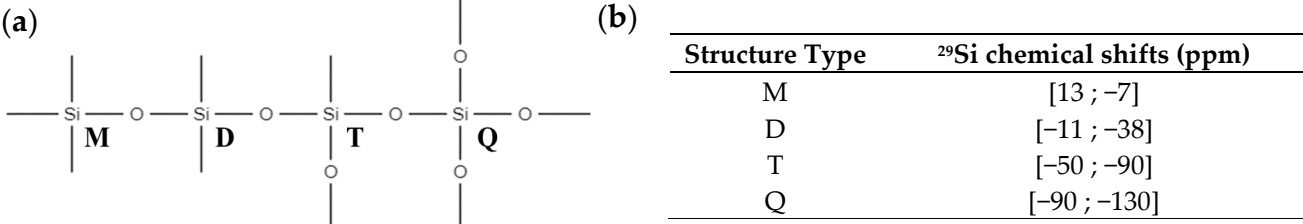

**(b)**

| Structure Type | $^{29}$Si chemical shifts (ppm) |
| --- | --- |
| M | [13 ; −7] |
| D | [−11 ; −38] |
| T | [−50 ; −90] |
| Q | [−90 ; −130] |

**Figure 1.** (**a**) Possible structures of silicon atoms in polysiloxanes and (**b**) corresponding chemical shifts of silicon atoms of a polysiloxane chain, depending on the silicon functionality, in $^{29}$Si NMR spectroscopy [25]. Chemical shifts are expressed in parts per million (ppm) and correspond to the relative difference in resonance frequency between the silicon atoms of the sample and tetramethylsilane (TMS).

NMR analyses of proton ($^{1}$H), carbon ($^{13}$C) and silicon ($^{29}$Si) were performed on the Silres H62C polysiloxane, using Bruker Avance III 400 MHz and 500 MHz spectrometers (Bruker, Germany). The spectrometers are equipped with 5 mm dual-channel probes (H-X). All the analyses were performed after dissolving the Silres H62C at 2 g·mL$^{-1}$ in deuterated chloroform (CDCl$_3$) purchased from Sigma Aldrich (Saint-Quentin-Fallavier, France) which contained 1% $v/v$ tetramethylsilane (TMS).

Quantitative $^{1}$H NMR analyses were carried out using a D1 relaxation time of 3 s and 100 scans. Qualitative $^{13}$C NMR analyses were carried out using a D1 relaxation time of 5 s and 1024 scans. A distortionless enhancement by polarization transfer experiment at 135° (DEPT-135) was also applied (D1 = 2 s, 512 scans) to identify the different carbon origins (CH and CH$_3$ positive, CH$_2$ negative, C$_{quaternary}$ zero). Heteronuclear single quantum correlation (HSQC, 1J correlations) and heteronuclear multiple bond correlation (HMBC, 2J to 4J correlations) experiments were used to obtain 2D $^{1}$H-$^{13}$C NMR spectra.

Quantitative $^{29}$Si NMR analyses were performed using a D1 time of 225 s, with a maximum T1 relaxation time of 50 s that was experimentally optimized. A sequence including a spin echo was applied to mask the contribution of the quartz of the NMR tube. The principle is based on the relaxation of the solid phase which is much faster than the liquid one. A D$_{16}$ delay of 15 ms before a 180° rotation of the spins, followed by a second D$_{16}$ delay of 15 ms before any acquisition, allows the removal of the contribution of the relaxing spins. HSQC and HMBC sequences were also used to obtain $^{1}$H-$^{29}$Si 2D NMR spectra, using a 1J Si-H coupling constant of 250 Hz.

### 2.3. Characterizations of Cross-Linked Silres H62C

The cross-linking temperature of Silres H62C was determined using differential scanning calorimetry (DSC) with a 404F1 calorimeter (Netzsch, Germany), in helium atmosphere, from 25 °C to 400 °C, with a heating rate set to 15 K/min.

Fourier transform infraRed (FT-IR) spectroscopy analyses were carried out on raw and thermally cross-linked Silres H62C samples to complete NMR characterizations. FT-IR spectra were recorded using a Vertex 70 spectrometer (Bruker, Germany) equipped with a He-Ne laser as the excitation source and an attenuated total reflectance (ATR) device equipped with a germanium crystal. Each spectrum was obtained from 64 scans, in the range of 4000–650 cm$^{-1}$ with a resolution of 2 cm$^{-1}$. The data were normalized by the

band at 2960 cm$^{-1}$ corresponding to the vibration of the C-H bonds in the CH$_3$ group. It was considered that this group does not evolve during cross-linking.

A thermogravimetric analysis (TGA) of the raw and thermally cross-linked Silres H62C was performed to investigate its behavior during its conversion into ceramics. The weight loss as a function of the temperature was recorded with STA 449 F1 TGA apparatus (Netzsch, Germany). For each analysis, 20 mg of polysiloxane was heated from room temperature up to 1000 °C at a heating rate of 1 K/min, then a dwell time of 1 h was applied. The ceramic yield ($\eta$) was expressed from the relative mass loss ($\Delta m$) between the initial mass of the polymer and the final mass of the ceramic (Equation (1)).

$$\eta \ (\%) = 100\% - \Delta m \ (\%) \tag{1}$$

STA 449 F1 TGA equipment (Netzsch, Germany) coupled with a QMS 403C mass spectrometer (Netzsch, Germany) was also used to analyze the released gases during the thermal decomposition associated with the ceramic conversion of the Silres H62C. Mass spectrometry is an ion separation technique, allowing the identification of the nature of gaseous species by measuring their $m/z$ ratio, corresponding to the ratio between the molar mass and the charge of the molecular ion. For TGA-MS experiments, a heating rate of 10 K/min was applied up to 1000 °C in argon.

The cross-linked ceramic precursor was also pyrolyzed at 1000, 1400 or 1700 °C for 1 h in argon with a heating rate of 1 K/min. An alumina tube furnace or a graphite furnace (ECM—Lilliput W resistor, France) was used to perform the thermal treatments up to 1400 or 1700 °C, respectively. The crystallization behaviors of the pyrolyzed ceramics were monitored using X-ray diffraction (XRD) with a D8 Advance diffractometer (Bruker, Germany) equipped with a LynxEye detector with Cu-K$\alpha$ radiation and for 2θ values ranging from 10 to 90° (step size of 0.05°).

## 3. Results and Discussion

### 3.1. Identification of the H62C Structure Using NMR Spectroscopy

The quantitative $^{29}$Si NMR analysis of the Silres H62C showed the presence of mono-functional (M units, 21.9%), di-functional (D units, 31.4%) and tri-functional (T units, 46.7%) silicon atoms (Figure 2). Interestingly, no Q unit was observed. Although the large proportions of D and M units are usually characteristics of a relatively linear polymer, the preponderance of T units showed that the H62C was also significantly branched. A unique population of mono-functional silicon atoms was observed at 10.2 ppm, indicating that the M units of the Silres H62C all showed the same structure. On the other hand, three distinct populations of D unit signals were observed at −31.9 ppm (D1), −34.0 ppm (D2) and −36.2 ppm (D3), respectively (Figure 2). According to the literature [25], the three populations could be attributed to di-functional Si atoms bound to the methyl and hydrogen groups, as they are all located around −35 ppm. Regarding the T units, the main signal centered at −78.9 ppm (T2) could be attributed to tri-functional Si atoms bound to the phenyl groups [25]. A second minority population of T units, representing 19.7% of the total T units, with a complex signal positioned between −63 and −73 ppm, was also observed (T1) and could correspond to Si atoms bound to non-aromatic substituents.

Quantitative $^1$H NMR spectroscopy of Silres H62C was also used to highlight the nature and concentration of the functional groups (Figure 3). Six different broad signals, noted from Ha to Hf, were centered at 0.1, 1.2, 3.8, 4.8, 5.9 and 7.4 ppm, respectively. Interestingly, the signals Ha and Hf were the main contributors and represented 49.4 and 37.2% of the total amount of protons in the polymer, respectively. The signal Ha corresponded to the protons of the methyl groups (-C**H**$_3$), whereas the signal Hf was first assigned to the aromatic protons. The signals Hb to He were related to the other functional groups of Silres H62C. In particular, the expected protons were those of the vinyl groups, with two environments (-C**H**=CH$_2$ and -CH=C**H**$_2$) and the silyl groups (Si-**H**).

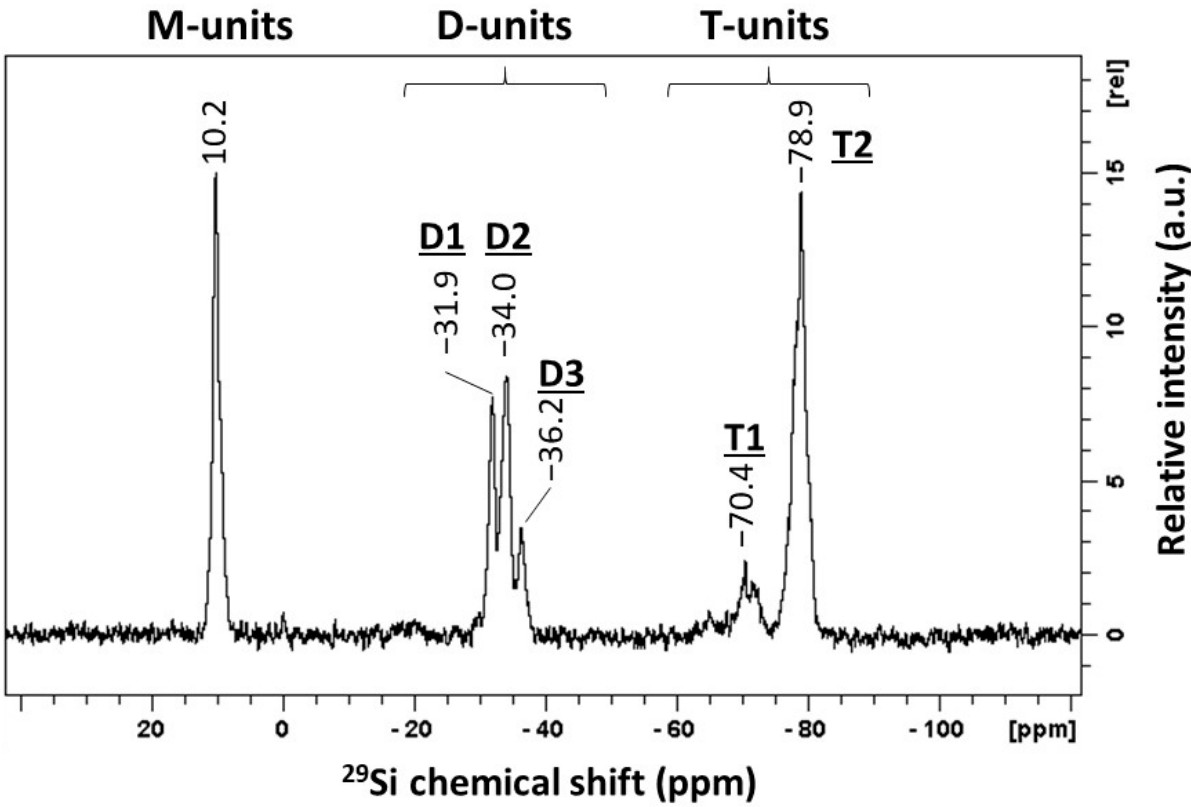

**Figure 2.** Quantitative $^{29}$Si NMR spectrum of Silres H62C polysiloxane dissolved at 2 g·mL$^{-1}$ in CDCl$_3$, with a relaxation time of 225 s and from 128 successive scans with spin echo acquisition.

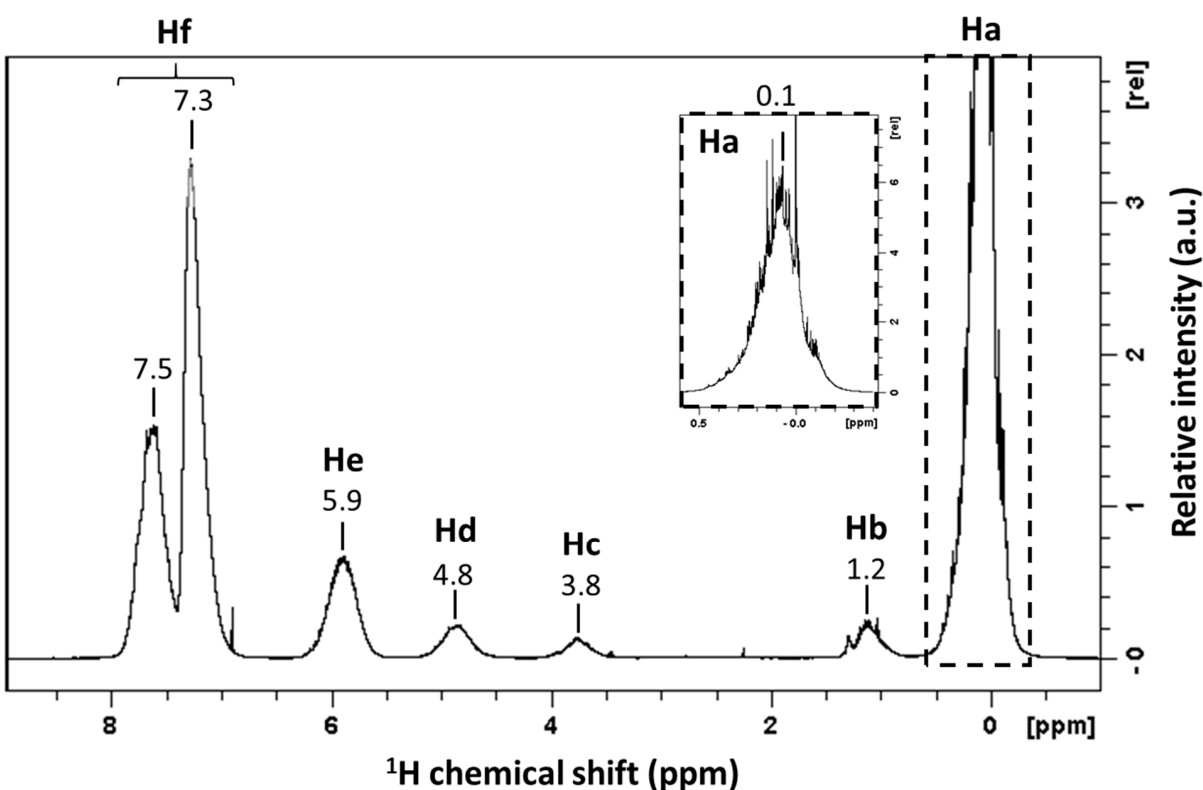

**Figure 3.** Quantitative $^1$H NMR analysis of Silres H62C polysiloxane dissolved at 2 g·mL$^{-1}$ in CDCl$_3$, with a relaxation time of 3 s and from 100 successive scans.

The direct [13]C and DEPT-135 (Figure 4) and HSQC [1]H-[13]C experiments (Figure 5) were performed to study further the attribution of the different functional groups of polysiloxane. Three domains were observed on the [13]C NMR spectrum. The first domain, between 125 and 138 ppm, corresponded to the sp[2] carbons. The peaks centered at 127.6, 130.0 and 134.0 ppm, respectively, which are positive in the DEPT-135, match with carbons having a double bond and linked to a single hydrogen atom (-CH=). These peaks were attributed to the carbons belonging to the aromatic (127.6 and 130 ppm) and vinyl groups (-CH=CH$_2$, 134 ppm), respectively. The HSQC experiment showed a clear correlation between the aromatic protons (Hf) and aromatic carbons at 127.6 and 130 ppm (Figure 5), in good agreement with this attribution. On the other hand, the carbon signal at 134 ppm and attributed to the -CH= of vinyl groups was also correlated to the Hf signal (Figure 5). Therefore, the Hf signal was supposed to be a superposition of the signals from the aromatic and vinyl protons and could not be used to directly quantify the phenyl groups in the Silres H62C polysiloxane. Interestingly, the DEPT-135 experiment also showed the presence of a =CH$_2$ groups in the first domain (125–138 ppm, Figure 4). This negative peak, centered at 133.6 ppm, was attributed to the second carbon of the vinyl groups (-CH=CH$_2$). Based on the corresponding correlation in the HSQC experiment (Figure 5), the He signal, centered at 5.9 ppm on the [1]H spectrum and representing 7.2% of the total proton amount, was then attributed to the =C<u>H</u>$_2$ protons of the vinyl groups. Consequently, the –C<u>H</u>= protons of the vinyl and aromatic groups, both superposed in the Hf signal, were estimated to represent 3.6 and 33.6% of the material protons, respectively.

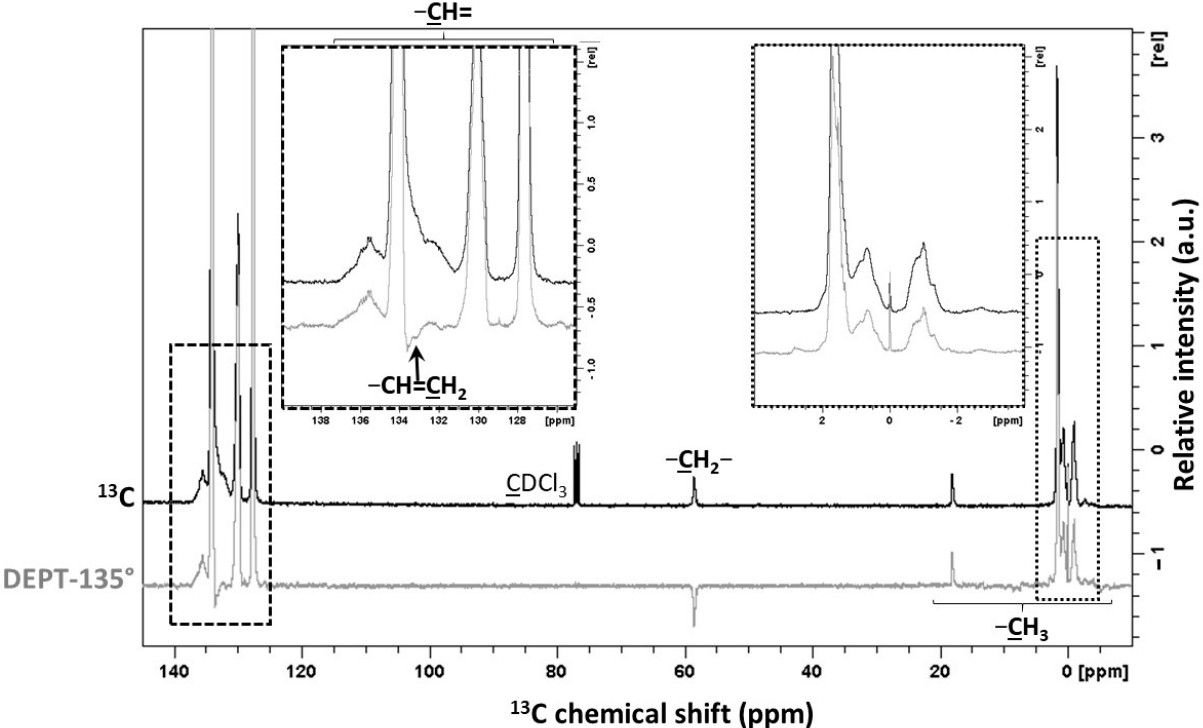

**Figure 4.** Direct [13]C NMR (black) and DEPT-135 (grey) experiment measuring Silres H62C polysiloxane at 2 g·mL$^{-1}$ in CDCl$_3$, with a relaxation time of 2 s and from 1024 successive scans.

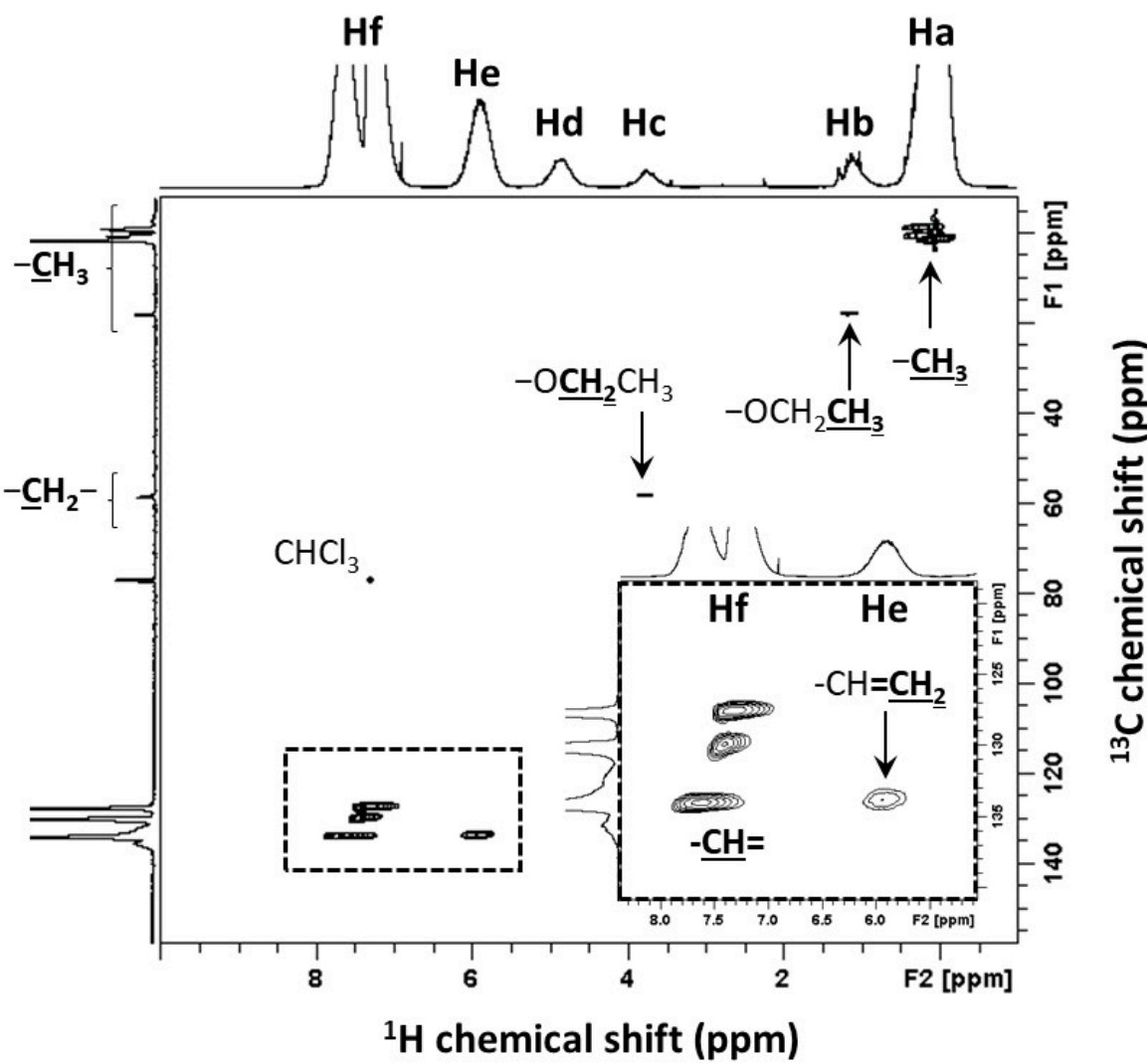

**Figure 5.** HSQC $^1$H-$^{13}$C NMR experiment measuring Silres H62C polysiloxane at 2 g·mL$^{-1}$ in CDCl$_3$, with a relaxation time of 1.5 s.

Others carbon atoms were identified on the $^{13}$C spectrum. In particular, a peak at 58.6 ppm, associated with a negative signal in the DEPT-135 experiment, was attributed to the carbon of a methylene bridge (-**C**H$_2$-). However, the high chemical shift of this methylene bridge indicated the presence of a deshielding heteroatom close to the methylene. The presence of alcoxy groups in the Silres H62C was unexpected as this polysiloxane is not supposed to cross-link through a mechanism based on hydrolysis/condensation. Nevertheless, the hypothesis of the ethoxy groups (-OCH$_2$CH$_3$) on silicon atoms was confirmed by comparing the chemical shifts of the corresponding protons (Hb: 1.15 ppm, Hc: 3.76 ppm) and carbons (18.2, 58.6 ppm) with the ones of tetraethyl orthosilicate (TEOS, $^1$H signals: 1.1–1.2, 3.8–3.9 ppm, $^{13}$C signals: 18.4, 60.2 ppm) [26–28].

Finally, the $^{13}$C and DEPT-135 experiments also showed the presence of several -**C**H$_3$ signals at 18.2, 1.7, 0.7 and −1 ppm, respectively (Figure 4). The first peak at 18.2 ppm and the corresponding Hb signal measured in $^1$H NMR (Figure 5) were attributed to the carbon and hydrogen atoms of the Si-O-CH$_2$-**C**H$_3$ structure, respectively. The three others carbon peaks between −1 and 2 ppm were correlated to the Ha signal and could correspond to various Si-**C**H$_3$ environments (Figure 5). Interestingly, the Hd signal (4.8 ppm), representing 2.5% of the total proton amount, was not correlated to any carbon atom and could, therefore, correspond to the silyl protons (Si-**H**).

The HSQC [1]H-[29]Si experiments were also performed to determine the first order coupling ([1]J) correlations between the protons and silicon atoms (Figure 6). Such correlations usually allow the identification of only the silyl (Si-H) positions on a polysiloxane chain. Very interestingly, a unique correlation was observed between the unattributed Hd signal (4.8 ppm) and the peaks corresponding to the di-functional silicon atoms (D, [−36.2; −31.9 ppm]), indicating that the silyl groups are only carried by Si atoms in the D configuration. Clearly, this observation also helped us to attribute the Hd signal to the silyl protons and to understand the absence of correlation between the Hd signal and carbon atoms (Figure 5). The HMBC [1]H-[29]Si experiments allow the characterization of the long-range correlation signals between protons and silicon atoms (from [2]J to [4]J, i.e., separated by two to four bonds), by suppressing the first-order correlation signals ([1]J observed in HSQC). This NMR technic was also used to characterize the Silres H62C polysiloxane (Figure 6). First, a clear correlation between the methyl proton signal (Ha, 0.1 ppm) and the mono-functional Si units (M, 10.2 ppm) was observed. Interestingly, no other correlation was observed with the M units, indicating that the polymer chain-ends are exclusively the trimethyl silane groups. On the other hand, the methyl protons (Ha) were also correlated to the di-functional Si units (D, [−36.2; −31.9 ppm]) but not to the tri-functional ones (T, [−78.9; −70.4 ppm]). Therefore, the methyl groups can only be found on the M and D siloxane units. Concerning the Si T units, two long-range correlations were only observed with the Hf protons (7.4 ppm). With the Hf signal being a superposition of the aromatic and vinyl protons (–C**H**=), it explains the presence of two Si T units: the main T signal positioned at −78.9 ppm (T2, 80.3% of the T units) corresponds to the Si atoms carrying a phenyl group in accordance with the literature [29], whereas the other T signal at −70.4 ppm (T1, 19.7%) could correspond to the Si atoms carrying a vinyl group. Lastly, no [1]H-[29]Si correlation was measured, neither with the HSQC nor HMBC sequences, for the Hb and Hc signals, confirming the respective attributions to the methyl and methylene protons of ethoxy groups. Indeed, intermediary oxygen atoms do not allow long-range correlation. Consequently, the NMR methodology used was sufficient to establish the presence of the ethoxy groups but not enough to identify their position on the polysiloxane chains at this point. Therefore, the ethoxy groups could be either on the D or T units.

The attribution of all of the H broad signals then allowed the quantification of the functional groups in the Silres H62C polysiloxane compound, knowing the number of protons for each group (Table 1). The Silres H62C constituent was found to be mainly composed of methyl (54.8%) and phenyl (22.4%) groups. The polysiloxane was also composed of 12 and 8.4% of the vinyl and silyl groups, respectively. Silres H62C is known to cross-link by hydrosilylation and, consequently, contains an excess of 43% of the vinyl groups. Interestingly, the polymer also contains ethoxy groups (2.4%). These unexpected alcoxy groups represented 29% of the silyls groups. This non-negligible proportion might allow the H62C cross-linking by a hydrolyze/condensation mechanism.

**Table 1.** Attribution of the [1]H NMR signals and quantification of the corresponding functional groups.

| Signals | Chemical Shift (ppm) | Attribution | Proportion of Protons (%) | Protons per Functional Groups | Proportion of Functional Groups (%) |
|---|---|---|---|---|---|
| Ha | 0.1 | Si-**CH**$_3$ | 49.4 | 3 | 54.8 |
| Hb | 1.2 | Si-O-CH$_2$-**CH**$_3$ | 2.3 | 3 ⎫ | |
| Hc | 3.8 | Si-O-**CH**$_2$-CH$_3$ | 1.4 | 2 ⎬ → | 2.4 |
| Hd | 4.8 | Si-**H** | 2.5 | 1 | 8.4 |
| He | 5.9 | Si-CH=**CH**$_2$ | 7.2 | 2 ⎫ | |
| Hf$_1$ | 7.3 | Si-**CH**=CH$_2$ | 3.6 | 1 ⎬ → | 12.0 |
| Hf$_2$ | [7.3; 7.5] | Si-**Ph** | 33.6 | 5 | 22.4 |
| Hf | [7.3; 7.5] | ∑(-C**H**)= (Hf$_1$ + Hf$_2$) | 37.2 | - | - |

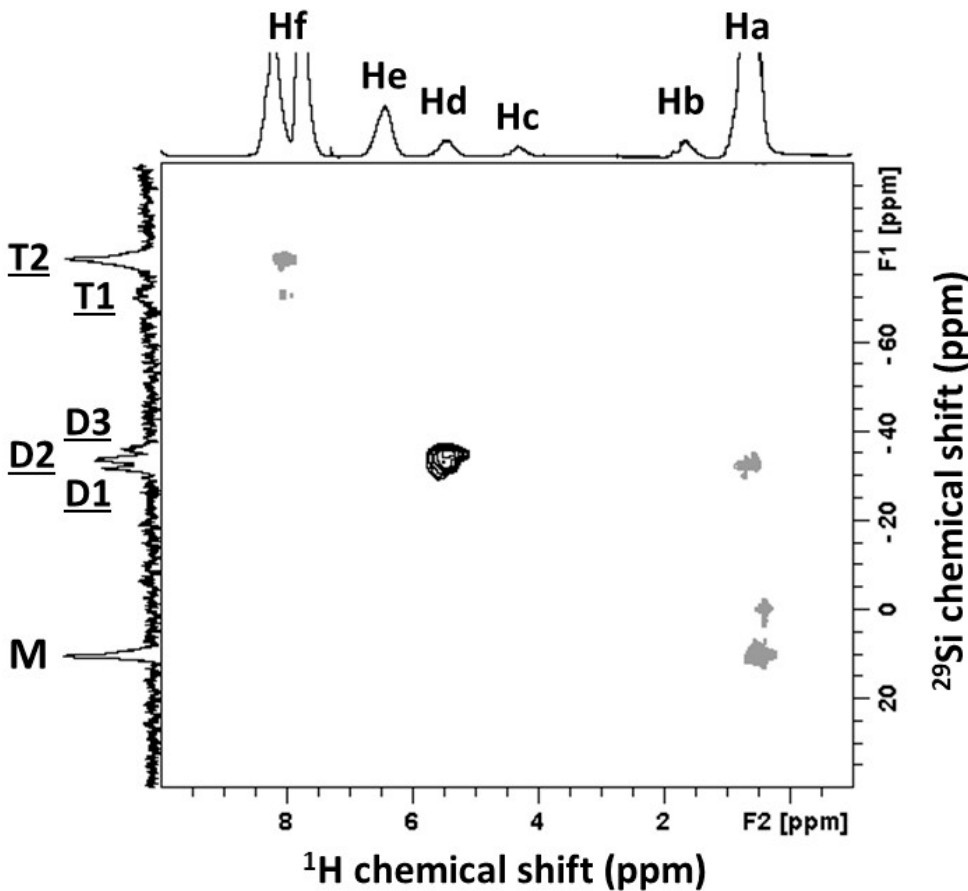

**Figure 6.** HSQC (black) and HMBC (grey) [1]H-[29]Si NMR experiments measuring Silres H62C polysiloxane at 2 g·mL$^{-1}$ in CDCl$_3$, with a relaxation time D1 of 2 s.

Considering the number of functional groups on each Si atom of the polysiloxane (Table 2), the position of each functional group on the polymer structure was highlighted (Table 3). The methyl groups were the first to be identified. Indeed, the 2D [29]Si-[1]H experiments showed that the Si M units (21.9% of Si atoms, corresponding to 37.5% of the functional groups) are only linked to the methyl groups (Figure 6). Moreover, the methyl groups (54.8%) were found to be carried by the Si M and D units. Consequently, the methyl groups carried by the M (68.4%) and D units (31.6%) were found to represent 37.5 and 17.3% of the total functional groups, respectively. On the other hand, the Si T units represent 26.6% of the functional groups (Table 3) and are linked to the phenyl and vinyl groups only (Figure 6). Considering that the phenyl groups (22.4%) are only carried by the Si T units, 4.2% of the functional groups were found to be vinyls carried by Si T units. The remaining vinyl groups (7.8% of the functional groups) are consequently carried by Si D units. Finally, considering the position of the previous functional groups, the silyls (8.4%) and ethoxy (2.4%) groups were found to be carried by the Si M units only, as already showed by the 2D [29]Si-[1]H experiments for the silyl groups.

Finally, after quantifying and positioning all of the functional groups present in the H62C polysiloxane, an example of the chemical structure was proposed for the polymer (Figure 7). Indeed, the proportions of Si atoms in the polymer (M: 21.9%, D: 31.4%, T: 46.7%), gave us a minimal atomic ratio of 7:10:15 (M:D:T) between the Si atoms. This ratio was found to be composed of minimal whole numbers, allowing the polymer to be liquid at room temperature. Obviously, this calculated structure (1207 g·mol$^{-1}$) was based on a unique chemical structure, but a mix of several lighter polysiloxane structure could also be considered. Nevertheless, the 7:10:15 ratio indicated that the polymer probably has a branched ladder-like structure.

**Table 2.** Attribution of the $^{29}$Si NMR signals and quantification of the functional groups carried by the corresponding Si atoms.

| Signals | Chemical Shift (ppm) | Attribution | Proportion of Si Atoms (%) | Functional Groups per Si Atoms | Proportion of Functional Groups (%) |
|---------|---------------------|-------------|----------------------------|--------------------------------|--------------------------------------|
| M | 10.2 | Mono-functional Si atoms | 21.9 | 3 | 37.5 |
| D | [−36.2; −31.9] | Di-functional Si atoms | 31.4 | 2 | 35.9 |
| T | [−78.9; −70.4] | Tri-functional Si atoms | 46.7 | 1 | 26.6 |

**Table 3.** Position of the functional groups on the H62C polysiloxane chain.

| | | | Si Atoms | | |
|---|---|---|---|---|---|
| | | | M Units | D Units | T Units |
| | | Proportion of Functional Groups (%) | 37.5 | 35.9 | 26.6 |
| Functional groups | Si-CH$_3$ | 54.8 | 37.5 | 17.3 | - |
| | Si-O-CH$_2$-CH$_3$ | 2.4 | - | 2.4 | - |
| | Si-H | 8.4 | - | 8.4 | - |
| | Si-CH=CH$_2$ | 12.0 | - | 7.8 | 4.2 |
| | Si-Ph | 22.4 | - | - | 22.4 |

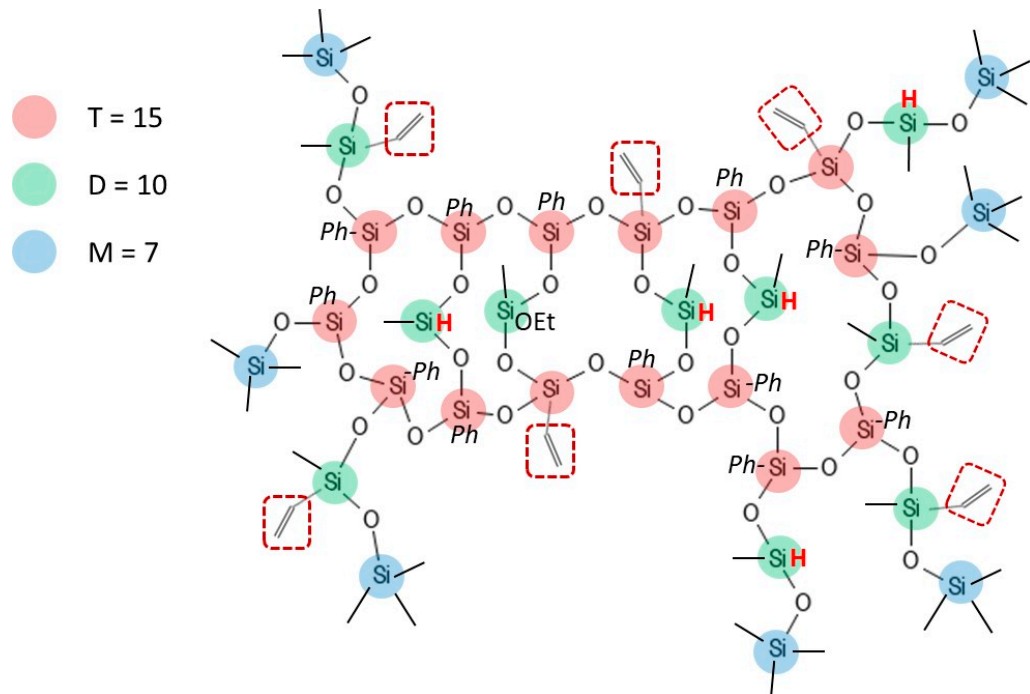

**Figure 7.** Proposed structure of Silres H62C, calculated from NMR characterization.

*3.2. Thermal Cross-Linking*

DSC analyses were carried out to investigate the cross-linking of the Silres H62C compound (Figure 8). The polysiloxane was used as received and after cross-linking at 200 °C for 1 h in air (CL_200). First, the DSC thermogram of the raw Silres H62C polymer showed a cross-linking peak spread between 171 and 242 °C, with an onset at 183 °C with a maximum centered at 190 °C and an enthalpy of cross-linking calculated to be 71.4 J·g$^{-1}$. After a thermal treatment at 200 °C, the subsequent DSC analysis of the cured polysiloxane remained flat, indicating that cross-linking was complete at this temperature.

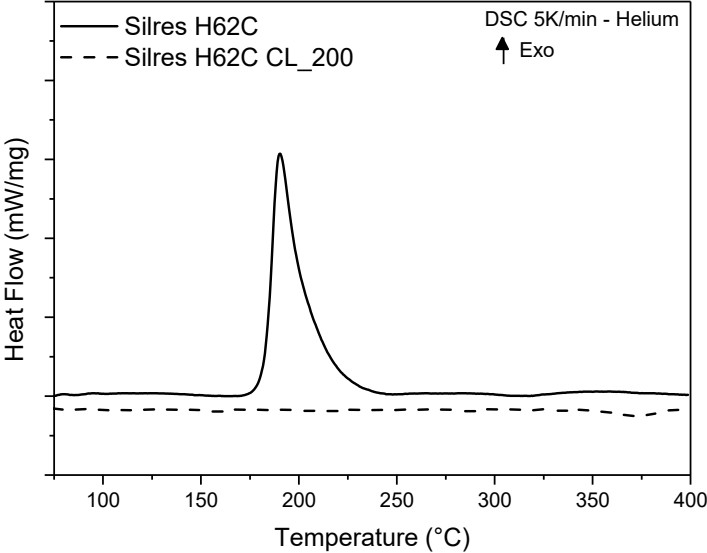

**Figure 8.** DSC analysis of starting and cross-linked H62C at 200 °C in air for 1 h, at a heating rate of 5 K/min under helium.

The structural evolution of Silres H62C during cross-linking was followed by FT-IR analyses (Figure 9). The assignment of each band is listed in Table 4. From the raw polysiloxane analysis (Figure 9a), a broad absorption band, between 990 and 1200 cm$^{-1}$, was related to the siloxane structure and corresponded to the various vibration modes of the Si-O bonds [30]. In particular, the very intense absorption band at 1053 cm$^{-1}$ was attributed to the asymmetric vibration of the Si-O bonds. On the other hand, the second band at 1134 cm$^{-1}$ indicates a branched polysiloxane structure with T units [30], in agreement with the NMR analyses.

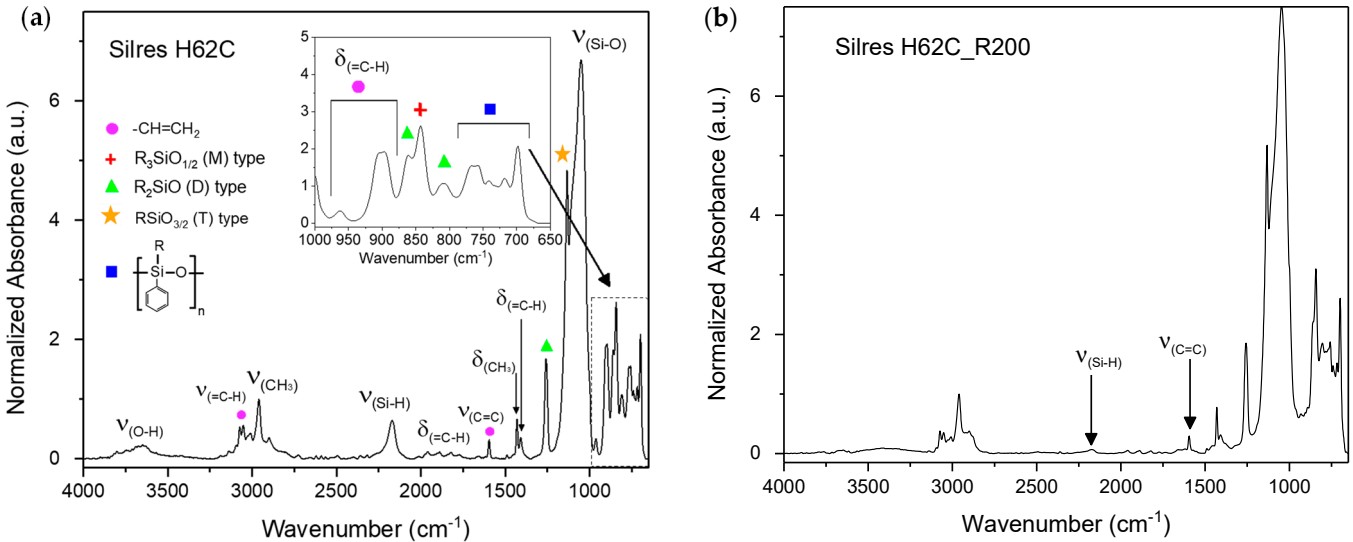

**Figure 9.** FT-IR/ATR analysis of (**a**) starting and (**b**) cross-linked Silres H62C at 200 °C in air for 1 h. Data normalized relatively to the C-H bonds in the CH$_3$ group at 2960 cm$^{-1}$. Figure (**a**) inset, zoom of the 650–1000 cm$^{-1}$ region.

**Table 4.** Attribution of major peaks resulting from FT-IR/ATR analyses of starting Silres H62C, attributions from [30].

| Wavenumber (cm$^{-1}$) | Normalized Absorbance | Mode |
|---|---|---|
| 3648 | 0.2 | $\nu$(O-H) |
| 3053 | 0.5 | $\nu$(=C-H) |
| 2960 | 1 | $\nu$(CH$_3$) |
| 2171 | 0.6 | $\nu$(Si-H) |
| 1959/1888/1822 | 0.1 | Harmonics of $\delta$(=C-H) |
| 1594 | 0.3 | $\nu$(C=C) |
| 1430 | 0.6 | $\delta$(CH$_3$) |
| 1407 | 0.3 | $\delta$(=C-H) in plane |
| 1260 | 1.67 | D unit polysiloxane (CH$_2$)$_2$SiO |
| 1134 | 4.8 | T unit polysiloxane (CH$_2$)SiO$_{3/2}$ |
| 1053 | 6.7 | $\nu$(Si-O) |
| 962/906/896 | 0.3/1.9/1.9 | $\delta$(=C-H) out of plane |
| 862 | 1.8 | D unit polysiloxane (CH$_2$)$_2$SiO |
| 842 | 2.6 | M unit polysiloxane (CH$_3$)$_3$SiO$_{1/2}$ |
| 809 | 1.0 | D unit polysiloxane (CH$_2$)$_2$SiO |
| 767/757/740/717/698 | 1.5/1.5/1.1/1.2/2.0 | D unit (Ph)(CH$_3$)SiO |

The peaks in the 698–767 cm$^{-1}$ range confirmed the presence of the aromatic groups. The bands at 1597 cm$^{-1}$ (stretching of **C=C** double bond) and 3052 cm$^{-1}$ (stretching of the =**C-H** bond) were attributed to the vinyl groups. The silyl functions were highlighted by the symmetrical vibration of the Si-H bond (2171 cm$^{-1}$) [30]. The association of the bands at 1260, 862 and 809 cm$^{-1}$ corresponded to the presence of the D units, whereas the bands at 1134 and 842 cm$^{-1}$ were attributed to the T and M units, respectively. The bands between 767 and 698 cm$^{-1}$ were attributed to the D units containing the phenyl groups. Furthermore, the large band centered at 3648 cm$^{-1}$ corresponds to the stretching of the O-H bond and can be assigned to the presence of water in the polymer and/or alcohol functions. The NMR results showed the presence of the ethoxy groups. These groups can be hydrolyzed, resulting in the presence of alcohol functions, conducting most likely to condensation reactions, thus generating water.

A few differences were observed on the FT-IR spectrum of Silres H62C after cross-linking for 1 h at 200 °C in air (Figure 9b). In particular, the substantial attenuation of the silyl band at 2171 cm$^{-1}$ (~94%) was characteristic of the polysiloxane cross-linking by hydrosilylation. The consumption rate of the silyl bands is detailed below. The vinyl absorption bands were slightly affected, particularly those at 1594 cm$^{-1}$. The presence of the remaining vinyl functions after cross-linking was in good agreement with the NMR results, showing an excess of vinyls compared to silyl groups (Table 3). However, the evolution of the vibrational band of the vinyl group at 1594 cm$^{-1}$ is not significant to enable a consumption rate determination. It could have been interesting to compare the consumption rate of the vinyl groups, measured using FT-IR, with the 43% of the excess calculated from the NMR (Table 1). Furthermore, the attenuation of the large band at 3648 cm$^{-1}$ may indicate that water has evaporated.

The evolution of the vibrational band of the silyl groups (2171 cm$^{-1}$) was then used to follow the cross-linking rate, as a function of the cross-linking time, at 200 °C in air (Figure 10). A fast consumption of the silyl functions was measured during the first minutes at 200 °C, with a consumption rate of 87% after only 10 min. After that, the cross-linking kinetics tended to decrease drastically to reach a consumption rate of 96.5% after 3 h. Then, a longer cross-linking time at 200 °C did not lead to a significant improvement in the cross-linking rate, probably due to a lack of mobility of the reactive species in the cross-linked

material. In fact, the Silres H62C resin contains a couple of platinum catalysts/inhibitors. The rapid thermal degradation of the inhibitor at 200 °C quickly triggers the catalysis of the cross-linking reaction. The catalyst then generates radicals that accelerate thermal cross-linking by hydrosilylation. A sufficient mobility of these radicals is important to cross-link the resin. When the cross-link density is too high, the mobility of the reactive species decreases, which highly slows down the reaction kinetic. Considering these results, a thermal treatment of 1 h at 200 °C in air was fixed for the following investigations. The polymer cross-linked under these conditions appeared translucent and rigid, and the cross-linking rate measured using FT-IR spectroscopy was 94.6%.

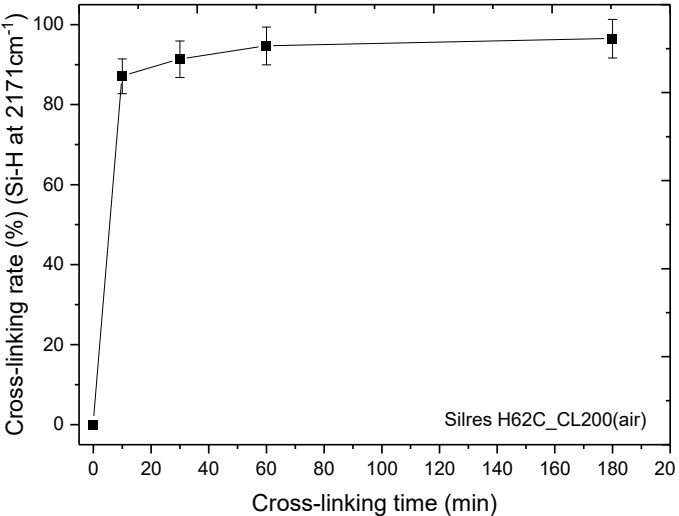

**Figure 10.** Cross-linking rate of the silyl functions (2171 cm$^{-1}$) of the Silres H62C polysiloxane as a function of cross-linking time at 200 °C in air.

### 3.3. Conversion Process: From the Polymer Precursor to Ceramic

The TGA analyses performed on Silres H62C showed a total weight loss of 27.5% occurring in four steps (Figure 11): between 150 and 270 °C (5.9%), from 270 to 350 °C (2.2%), from 350 to 550 °C (15.1%) and then from 550 to 1000 °C (4.3%). Interestingly, a weight loss of 2.6% was measured at 200 °C and was higher than the content of the volatile specified in the technical datasheet of Silres H62C which mentioned that it might not exceed 1.5% for a thermal treatment at 200 °C during 1 h. Silres H62C embeds a Pt complex catalyst and an inhibitor of which the contents are unknown. Dimethyl fumarate and dimethyl maleate are commonly used as inhibitors at a level of 50 ppm [31].

Cross-linking at 200 °C in air leads to a thermoset polymer that is stable up to 200 °C. The TGA analysis of the cross-linked Silres H62C lead to a total weight loss of 24.1% occurring in three steps, between 150 and 270 °C (1.0%), from 270 to 550 °C (18.3%) and then from 550 to 1000 °C (4.9%). Therefore, the ceramic yield is 72.5% for the pristine Silres H62C versus 73.9% for the cross-linked Silres H62C, taking into account the weight loss during cross-linking at 200 °C in air. A difference in the ceramic yield of 1.4 points is not significant to claim that cross-linking improves the ceramic yield as Silres H62C embeds a catalyst.

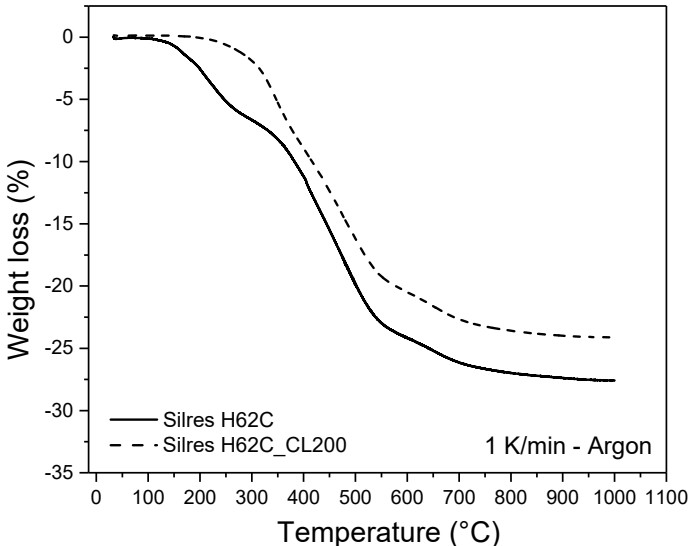

**Figure 11.** TGA analyses of starting and cross-linked Silres H62C. TGA was performed at 1 K/min up to 1000 °C for 1 h in argon.

The high weight losses for Silres H62C were attributed to the decomposition of the organic groups during ceramization. The TGA-MS analyses (Table 5) identified the presence of the hydrocarbons species (methane $m/z = 16$, vinyls $m/z = 27$, benzene $m/z = 78$), methoxy ($m/z = 31$), ethoxy ($m/z = 45$) and then dihydrogen ($m/z = 2$) release. Contrary to the ethoxy groups, the methoxy groups were not detected by the NMR spectroscopy. Consequently, the $m/z = 31$ signal was considered to be part of the fragmentation pathway of the ethoxy groups (base ion peak: $[CH_2OH]^+$). Water ($m/z = 18$) was not detected by MS under 300 °C meaning that the O-H bond detected by FT-IR/ATR (Figure 9) can be only attributed to alcohol functions and, therefore, to the potential presence of silanol groups (Si-OH) in the polymer structure. On the other hand, the condensation of these potential silanols could explained the water release at temperatures higher than 300 °C. The release of hydrocarbons (benzene, vinyl, methane) from 375 °C were in good agreement with the functional groups identified by NMR. Ethane (with a base ion peak at $m/z = 28$) was also measured and could be formed through the condensation reaction of Si-O-Et with Si-H (Equation (2)) [32]:

$$\equiv\!Si\text{-}OEt + H\text{-}Si\!\equiv \; \rightarrow \; \equiv\!Si\text{-}O\text{-}Si\!\equiv + C_2H_6 \tag{2}$$

**Table 5.** Degradation products and temperatures of apparition measured using TGA-MS for Silres H62C. Heating rate is fixed to 10 K/min from room temperature up to 1000 °C in argon.

| Products | Temperature (°C) | $m/z$ |
|:---:|:---:|:---:|
| $CH_3SiH$ | 300–700 | 44 |
| Water $H_2O$ | 350–650 | 18 |
| Ethoxy $C_2H_5O^-$ | 350–700 | 31, 45 |
| Ethane $C_2H_6$ : $[C_2H_4]^+$ | 350–700 | 28 |
| Benzene $C_6H_6$ | 375–650 | 78 |
| Vinyl $CH_2{=}CH$ | 400–650 | 27 |
| Methane $CH_4$ | 400–800 | 16 |
| Dihydrogen $H_2$ | 500–1000 | 2 |

Dihydrogen ($m/z = 2$) was detected from 500 °C up to 1000 °C. Bahloul-Hourlier et al. [32] have proposed various mechanisms to explain the formation of $H_2$ during the thermal conversion of polysiloxane. Indeed, dihydrogen could be formed either by the condensation of potential silanols with the residual silyl groups up to 650 °C (Equation (3)) or at higher

temperatures by the condensation of Si-CH$_3$ with the silyl groups (Equation (4)) or with other Si-CH$_3$ groups (Equation (5)) [33,34]:

$$\equiv\text{Si-OH} + \text{H-Si}\equiv \rightarrow \equiv\text{Si-O-Si}\equiv + \text{H}_2 \qquad (3)$$

$$\equiv\text{Si-CH}_3 + \text{H-Si}\equiv \rightarrow \equiv\text{Si-CH}_2\text{-Si}\equiv + \text{H}_2 \qquad (4)$$

$$\equiv\text{Si-CH}_3 + \text{CH}_3\text{-Si}\equiv \rightarrow \equiv\text{Si-CH}_2\text{-CH}_2\text{-Si}\equiv + \text{H}_2 \qquad (5)$$

The release of the silicon-based species, such as CH$_3$SiH ($m/z$ = 44), between 300 and 700 °C due to the redistribution reactions between the Si-O bonds and Si-H or Si-C bonds was also in good agreement with the literature [35–37].

XRD characterizations were performed on the Silres H62C resin cross-linked at 200 °C in air and pyrolyzed in argon at temperatures between 1000 and 1700 °C (Figure 12). For the pyrolysis temperatures of 1000 and 1200 °C, the obtained materials exhibited a diffractogram typical of an amorphous material, with a broad hump around centered around 2θ = 20°, associated with the amorphous Si-O bonds. A signal with a broad peak width for Silres H62C treated at 1400 °C was then observed, indicating the progressive apparition of a crystalline phase between 1200 and 1400 °C. Five distinct and fine peaks were finally observed for 35.6, 41.3, 59.9, 71.7 and 75.4° on the diffractogram of the materials pyrolyzed at 1700 °C. All diffraction peaks were associated with the cubic silicon carbide (3C-SiC) crystalline phase. The crystallite size calculated from the Scherrer equation using the (111) peak was 73 nm at 1700 °C.

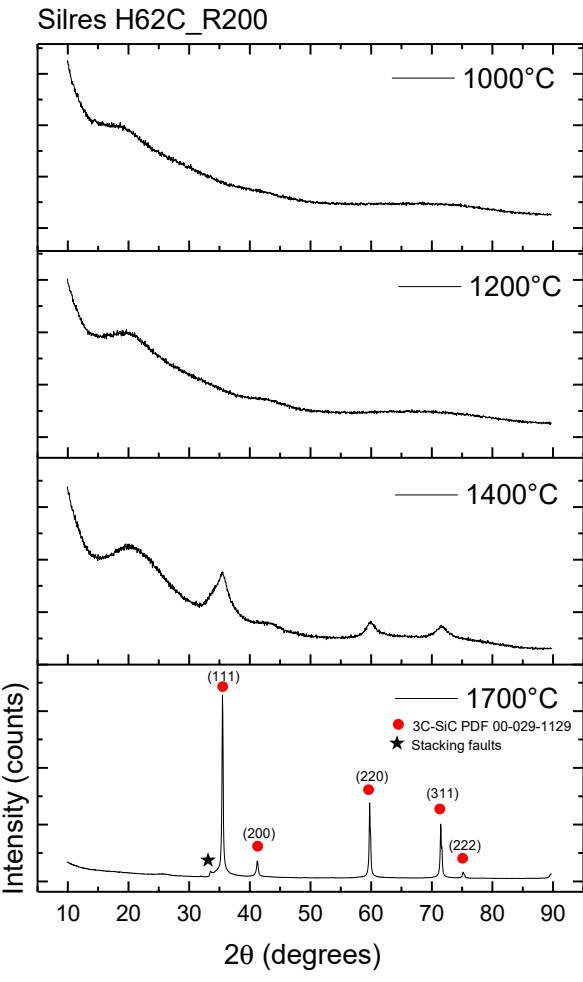

**Figure 12.** XRD analyzes of Silres H62C cross-linked at 200 °C and pyrolyzed at 1000, 1200, 1400 and 1700 °C in argon (heating rate set to 1 K/min, dwell time of 1 h).

In addition, a small peak to the left of the main peak was observed at 33.7°. This small peak could fit with the (100) plane of the 4H-SiC phase. However, the complementary diffraction peaks related to this phase were not observed. In the literature, a debate concerning the origin of this peak has been the subject of many papers. Pujar et al. showed, using a simulation procedure of the diffractograms, that stacking faults within the formed SiC grains could be responsible for its appearance [38].

Finally, an observation using transmission electron microscopy of the ceramic converted at 1700 °C reported in a previous work [21] confirmed the presence of the SiC crystalline phase and residual carbon-rich phase (Figure 13). The final composition of the ceramic obtained from the conversion of the Silres H62C polysiloxane resin at 1700 °C in argon gave a solid compound made of 82.15 wt% SiC, 0.69 wt% $SiO_2$ and 17.16 wt% of free C. The results obtained using the Scherrer equation from XRD give smaller-sized particles than the observation using TEM which shows an average particle size of 122 nm, accessed by image analysis. The discrimination of the particles using image analysis is made difficult by the superposition of particles and the presence of numerous twins.

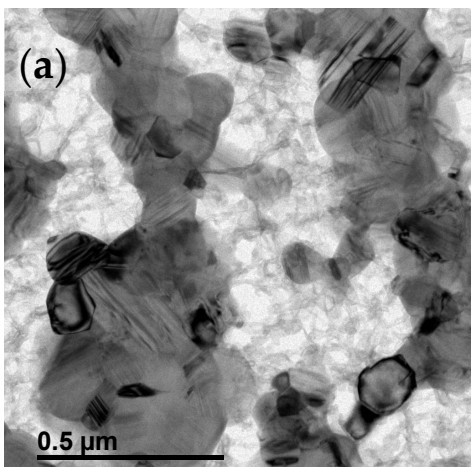 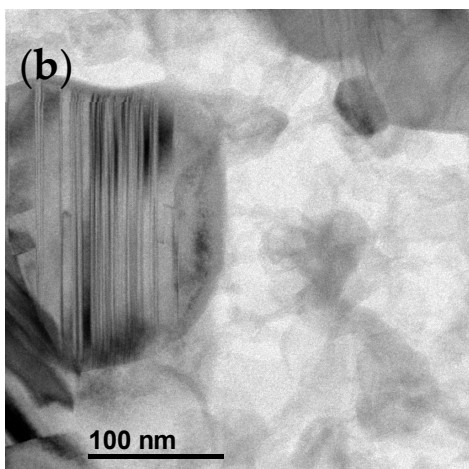

**Figure 13.** Reprinted/adapted from Vry et al. [21] (**a**,**b**) Transmission electron microscopy (TEM) view in bright field mode of the Silres H62C polysiloxane preceramic polymer sample cross-linked at 200 °C in air and pyrolyzed at 1700 °C in argon.

## 4. Conclusions

The polymer-derived ceramic Silres H62C can be described as a "ladder-like" polysiloxane with 21.9, 31.4 and 46.7% of mono-functional, di-functional and tri-functional silicon atoms, respectively. The NMR investigations identified methyl, phenyl, vinyl and silyl groups, as expected but also ethoxy functions. The quantification of the functions enabled the design of the molecular structure of the polysiloxane containing 8.4 and 12.0% of silyl vinyl functions, respectively. This molecular structure allows the polymer to be liquid at room temperature and to cross-link by hydrosilylation at a starting temperature of 183 °C and with an enthalpy of 17.4 J·g$^{-1}$. The cross-linking rate of the polysiloxane reached 94.6% after a first thermal treatment of 1h at 200 °C in air. The conversion at a high temperature of the cross-linked Silres H62C (1000 °C in argon) leads to a ceramic yield of 73.9%. The MS analysis highlighted that hydrocarbons and dihydrogen are released during the conversion of the polymer into ceramic. The hydrocarbon fragments are in agreement with the structure proposed by NMR. The resulting ceramic was amorphous at temperatures up to 1200 °C, and the first signs of crystallization were noticed at 1400 °C. At 1700 °C, the ceramic seemed to be mainly composed of the 3C-SiC phase. The TEM characterizations showed that the ceramic also contains a residual amorphous carbon phase. The determination of the chemical structure and investigation of the thermal behavior of the commercial Silres H62C compound gave an insight into the mechanisms involved in the conversion of

the polymer into ceramic and, therefore, help in better selecting SiC precursor candidates for 3D printing based on photopolymerization reactions.

**Author Contributions:** S.V. developed the conceptualization, formal analysis, methodology and writing (original draft). M.R. contributed to the conceptualization, methodology, writing (review and editing) and project administration. S.R. contributed to the formal analysis, methodology, software and writing (original draft). P.-A.B. contributed to the methodology and software. G.B.-G. contributed to the conceptualization, methodology, writing (review and editing) and project administration. All authors have read and agreed to the published version of the manuscript.

**Funding:** This research received no external funding.

**Institutional Review Board Statement:** Not applicable.

**Data Availability Statement:** Not applicable.

**Conflicts of Interest:** The authors declare no conflict of interest.

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
