# Peer review of "Silicon Carbide Precursor: Structure Analysis and Thermal Behavior from Polymer Cross-Linking to Pyrolyzed Ceramics"

_ceramics, doi:10.3390/ceramics5040076_

Round 1

Reviewer 1 Report

The manuscript concerns the characterization of a commercial SiC preceramic polysiloxane composed of a variety of monofunctional, difunctional, trifunctional Si units bearing methyl, vinyl, phenyl and H functional groups.

Most of the manuscript is dedicated to the NMR characterization with a complete set of 1H, 13C and 29Si NMR experiments, which have been seriously conducted. The description and analysis of the spectra could have been shortened, and the authors could have better relied on chemical shift values found to literature data to assign the peaks. For examples, 1H chemical shift at 4.8 ppm could have been safely assigned to Si-H groups…. 29Si chemical shifts of T units excluded the presence of methyl groups bonded to these units (expected around -65 ppm)….  I do not think there was a need to calculate the chemical shift of ethoxy groups, instead of using the experimental one found for TEOS. 

The quantitative analysis of the various Si sites and organic functional groups is interesting and lead to a structural model (Fig. 7) that the authors qualify as “a relatively linear but highly branched ladder-like structure”. I did not find the linear aspect quite evident from the proposed model.

Then the authors give some information regarding the polymer-to-ceramic transformation, especially by performing TG-MS analyses. I am not a specialist of the analysis of the MS data, but I found some discrepancies between the assignments of the degradation products listed in table 6 and data from the literature (for example: J. Europ. Ceram. Soc. 25 (2005) 979–985; J. Sol-Gel Sci. and Tech. 26 (2003) 279–283). Are you sure that the fragment 31 is due to methoxy groups, and not part of the fragmentation pathway of ethoxy groups? Also, are SiO, CO or CO2 expected to form at so low temperatures? The fragment 28 was assigned in the cited papers to C2H6 that can form by reaction between Si-H and Si-OEt. Are the authors sure that O2 can form by Si-O cleavage in this temperature range, and thus be responsible to combustion reactions under Ar?  

I really recommend that authors check their attributions and reinforce them with references (which are currently lacking), and better discuss the thermal behavior of this preceramic polymer. This part weakens the purpose of this manuscript.

Finally, I do not find it necessary to present TEM images that have already been published and which, for me, are outside the scope of this manuscript. 

Other comments: 

- Systematic message of errors to recall the figures and tables in the text. 

- The term "clusters" used for the 1H NMR spectrum seems inappropriate to me. I would have used broad signals instead.

- I suggest to modify the title: Thermal behavior from crosslinked polymer to pyrolyzed ceramics

Author Response

Please see attached the reply of your review report.

Best regards,

Sébastien Vry

Reviewer 2 Report

This paper concerns the determination of the chemical structure of the Silres H62C polymer by nuclear magnetic resonance (NMR) spectroscopy,and the crosslinking and polymer-ceramic conversion by FTIR, TG, MS  and XRD. The results are detailed and convincing. 

Author Response

Please see attached the response of your review report.

Best regards,

Sébastien Vry

Reviewer 3 Report

Manuscript ID (ceramics-1938587):

Sébastien Vry et al.

Authors of paper investigated the chemical structure of the Silres H62C polymer by NMR spectroscopy. DSC used to characterize to Silres H62C cross-linking temperature, cross-linking reaction is followed by FTIR-spectroscopy. TGA was used to determine the yield of 3C-SiC, which is found to be >80%.  Finally, XRD and TEM were used to analyze the phases and microstructure. Overall, the manuscript is promising and can be published in this journal after revising with below comment.  

Comment:

I suggest authors to measure the crystallite size from the analyzed XRD (Fig. 12) at 1700oC and SiC crystalline size from TEM images of Fig. 13. This could give readers or industries to scale-up powder synthesis of 3C-SiC using this methodology.

Author Response

(The authors gave the same response as above.)

Round 2

Reviewer 1 Report

The authors have responded well to my comments and I recommend publication in its current form.